# Breakdown of Symbiosis in Radiation-Induced Oral Mucositis

**DOI:** 10.3390/jof7040290

**Published:** 2021-04-12

**Authors:** Gianluca Ingrosso, Simonetta Saldi, Simona Marani, Alicia Y. W. Wong, Matteo Bertelli, Cynthia Aristei, Teresa Zelante

**Affiliations:** 1Radiation Oncology Section, Department of Medicine and Surgery, University of Perugia, 06129 Perugia, Italy; gianluca.ingrosso@unipg.it (G.I.); maranisimona@libero.it (S.M.); cynthia.aristei@unipg.it (C.A.); 2Radiation Oncology Section, Perugia General Hospital, 06129 Perugia, Italy; saldisimonetta@gmail.com; 3Division of Clinical Microbiology, Department of Laboratory Medicine, Karolinska Institute, 141 86 Stockholm, Sweden; alicia.wong@ki.se; 4MAGI Group, 25010 San Felice del Benaco (BS), Italy; matteo.bertelli@assomagi.org; 5Pathology Section, Department of Medicine and Surgery, University of Perugia, 06129 Perugia, Italy

**Keywords:** oral mucositis, *Candida albicans*, microbiota, fungal infections, lactobacilli, radiation therapy

## Abstract

Oral mucositis is an acute side effect of radiation therapy that is especially common with head and neck cancer treatment. In recent years, several studies have revealed the predisposing factors for mucositis, leading to the pre-treatment of patients to deter the development of opportunistic oral fungal infections. Although many clinical protocols already advise the use of probiotics to counteract inflammation and fungal colonization, preclinical studies are needed to better delineate the mechanisms by which a host may acquire benefits via co-evolution with oral microbiota, probiotics, and fungal commensals, such as *Candida albicans*, especially during acute inflammation. Here, we review the current understanding of radiation therapy-dependent oral mucositis in terms of pathology, prevention, treatment, and related opportunistic infections, with a final focus on the oral microbiome and how it may be important for future therapy.

## 1. Radiation-Induced Oral Mucositis

In the treatment of head and neck (HN) cancers, radiation therapy (RT) plays a key role [1] and can be used as a radical or adjuvant treatment either alone or in concert with chemotherapy (CT) [1]. Although technological advances in RT have improved oncological outcomes [2,3,4], oral mucositis (OM) is still a common acute side effect [5].

Sonis et al. suggested that there are five stages of RT- and/or CT-induced OM: initiation, signaling, amplification, ulceration, and healing (Figure 1) [6]. Radiation-induced free radicals and DNA damage modify the intra- and inter-cellular signaling pathways that regulate epithelial and immune-cell proliferation, differentiation, and death [7]. The response cascade causes inflammation and activates apoptosis and epithelial hypoplasia. At the cumulative RT dose of 20 Gy (i.e., the threshold for mucosal tolerance), pro-inflammatory cytokines are released from the vascular epithelium and connective tissue at the site of injury [8] and cause oral mucosa, tongue, and pharynx hyperemia and erythema [7]—the first visible signs of OM (Figure 1). Damage to the basement membrane beneath the epithelial cell layer leads to protective barrier loss, which determines the degree of desquamation and ulceration when the cumulative dose reaches 30 Gy [8]. This process then progresses, and confluent lesions appear by the fourth to fifth week of RT, eventually followed by ulceration, necrosis, and bleeding. Oral pain, odynophagia, reduced oral intake with subsequent weight loss, nutritional deficits, and fungal infections are the main clinical complications and can lead to the increased use of narcotic analgesics and prolonged hospitalization. This clinical scenario can result in treatment interruptions that can negatively affect disease control [8]. The core oral microbiome is modulated during RT [9,10]. After irradiation, there are alterations in the main phyla abundances of the core oral microbiome. In particular, a significant increase in Lactobacilli has been described, but no studies have been able to explain what exactly causes these changes (Figure 1) [9,11]. Furthermore, Gram-negative bacteria, such as Enterobacteriaceae spp., *Pseudomonas* spp., and *Escherichia coli*, and Gram-positive bacteria, such as *Staphylococcus* spp. and *Streptococcus* spp., as well as the fungus *Candida albicans*, have been isolated from oral swabs during OM post-RT [12,13]. Interestingly, xerostomia was associated to an increase of *Candida* spp. and *Lactobacilli* spp. [11].

Although yet unproven, it is highly likely that alterations in the composition of the individual oral microbiome represent an important risk factor in the predisposition to dysregulated inflammation in OM or secondary infections, but the precise mechanisms are still unclear.

Here, we review studies that have shown how RT may induce OM, the types of fungal infection that may occur, how oral microbiota may affect the risk of OM and concomitant fungal infections, and how the local activation of dysregulated immune responses potentially influences the risk of developing fungal infections.

## 2. Grading of OM

Studying the potential differences in the microbiome in relation to disease severity, and the beneficial effects of probiotics, is likely to provide insights into the causative or protective roles of bacteria in OM pathology.

In this context, OM pathology is defined by a variety of clinical scales available for OM grading. The Radiation Therapy Oncology Group (RTOG) scale, the most widely used, assesses and scores acute RT morbidity criteria for mucosal membranes [14]. The World Health Organization (WHO) Oral Toxicity scale measures OM anatomical, symptomatic, and functional elements, while the Western Consortium for Cancer Nursing Research scale only describes the anatomical changes associated with OM [15]. The OM Index (OMI) scores OM severity by grading erythema, ulceration, atrophy, and edema from 0 (none) to 3 (severe). The highly reproducible OM Assessment Scale (OMAS) allows for evaluation over-time and the objective evaluation of ulceration or pseudo-membrane presence and size (0 = no lesion; 1 = lesion of <1 cm^2^; 2 = lesion of 1–3 cm^2^; 3 = lesion > 3 cm^2^) and erythema (0 = none; 1 = not severe; 2 = severe) on the upper and lower lips, right and left cheeks, right and left ventral and lateral tongue, floor of the mouth, soft palate, and hard palate [16]. More recently, the Common Terminology Criteria for Adverse Events (CTCAE) scale, and its updated versions (the latest being v5), is becoming increasingly popular and is already widely used. CTCAE Grade I is characterized by mild pain or congestion and does not require analgesics. Grade II includes the development of patchy mucositis and serosanguineous discharge, which require analgesic treatment. In Grade III, the development of confluent mucositis or severe pain requires narcotic analgesics, while in Grade IV ulcers, necrosis or bleeding appear.

Considering the available studies, the change of the core microbiome is happening more conspicuously during the ulceration phase, where colonization of certain bacteria occurs and overall microbial diversity is reduced, as indicated by 16S sequencing technology or cultures (Figure 1) [10,17]. Also, in this phase, the occurrence of opportunistic infections is shown (Figure 1) [6]. The appearance of hyposalivation is also pivotal for *Candida* oral infections [18,19].

Overall, it seems that changes in the oral microbial community are correlated with the evolution and exacerbation of RT-induced OM [20]. In other words, as the oral mucosal lesions worsen from Grade 1 to Grade 4, researchers found that bacterial UniFrac distances between patients and controls increased significantly per grade level [20]. For example, a recent study showed that *Pseudomonas* and *Treponema* were positively associated with the dose of radiation applied. In addition, the peak abundances of *Treponema* frequently coincided in time with the emergence of Grade 3 and Grade 4 mucositis [10].

We will discuss more findings of studies that have investigated the oral microbiome and the effects of RT in terms of OM grade in Section 7.

## 3. Risk Factors for OM

Many risk factors are involved in OM development and intensity, and these include the RT-related factors: single- and total-dose, fractionation schedule, technique, tumor site, and treatment volume. Moreover, concurrent CT may contribute to earlier mucositis development and severity. Patient-related factors include alcohol intake and tobacco use, diet, poor oral hygiene, dental status, and the local microbial environment [21]. When Chen et al. analyzed the predictive factors for the prevalence of severe OM and OM-related symptoms [22], concurrent CT, higher RT dose, smoking, and a lower body mass index emerged as significant risk factors for developing severe OM, while concurrent RT-CT, RT dose, and smoking predicted OM-related symptoms [22].

Recently, it has been found that by using 16S rRNA gene sequencing, it is possible to assess that alteration of the oral core microbiome or dysbiosis may be considered a risk factor for OM in HN patients, since a particular microbial signature may exacerbate the severity of mucositis [10,20].

More interestingly, the diversity and the oral microbiome profile can be used for the prediction and thus prevention of the development of severe mucositis during RT [20]. A random forest model has been applied knowing the oral microbiome profile of HN patients undergoing RT versus controls. The model was predictive for the aggravation of mucositis from Grade 2 to Grade 3 or 4. Using a more appropriate cohort of patients would help to predict mucositis aggravations before RT and therefore to establish which kind of oral microbiome may represent a risk factor for the development of severe OM [20].

## 4. Opportunistic Fungal Infections in Patients Treated with RT

RT for HN cancers causes local tissue changes based on damage to basal epithelial cells and connective tissue, causing vascular permeability [23]. This impairment of the mucosal barrier, which is clinically detectable after 2–3 weeks of RT, can lead to patients experiencing toxicity events such as pain and the impaired oral intake of fluids and nutrients [23,24]. The combination of mucosal barrier impairment and radiation-related dysbiosis favors colonization and invasion by commensal bacteria and fungi. Dysbiosis clearly alters microbial homeostasis at the mucosal interface, leading to the hyperactivation of pattern recognition receptors (PRRs) and, thus, a predisposition to OM [12]. Thus, following barrier disruption, commensal overgrowth by activating PRRs favors ecosystem alteration and hyperactivation of the oral inflammatory response.

Radiation-induced tissue inflammation and cell apoptosis can develop from the constant mucosal inflammation caused by the treatment schedule, which is characterized by a daily dose-delivery for several weeks. This chronic and accumulated dose exposure results not only in impairment of the epithelial barrier but also in selective pressure on the microorganisms normally present in the oral cavity and pharynx, leading to the overgrowth of polymicrobial commensal dysbiotic bacterial communities [23,24].

For instance, during HN RT, the dynamic synchronous variations in the relative abundances of four genera of bacteria (*Fusobacterium*, *Porphyromonas*, *Treponema*, and *Prevotella*) are reportedly related to the onset of high-grade toxicity [10]. Eventually, the combination of mucosal barrier impairment, hyposalivation, and dysbiosis favors colonization and invasion by fungi (Figure 1) [19,25].

Fungal disease exacerbates the radiation-induced toxicity and may necessitate a significant modification of the RT regimen and even discontinuation of the treatment, compromising the efficacy of the cure [24]. Fungal infections are mainly due to commensal *Candida* species that normally reside on the oral mucosa and in the lumen of the gastrointestinal tract [13,23,24]. The main commensal yeast in the digestive mucosa is *Candida albicans*, which is responsible for more than 80% of HN infections, followed by *C. glabrata* and *C*. *tropicalis.* [13,25].

Fungal colonization is present in about 50% of HN cancer cases before RT begins, with the percentage rising to 75% during treatment [26]. From a pathophysiological point of view, fungal infections are related to a pH change in the microenvironment that favors the overexpression of fungal virulence factors, such as secreted aspartyl proteinase, a change in the composition of the saliva, and immune dysregulation (see Section 5).

In physiological conditions, mucins and proteoglycans in saliva bind to microorganisms, which are then eliminated from the mucosa during swallowing. Moreover, saliva contains antimicrobials, such as lysozyme, secretory antibodies (IgA), and lactoferrin that inhibit the epithelial adhesion of fungi to the mucosa [23,24,26]. During and after RT, the reduction in salivary flow and changes in salivary composition increase the acidity of the oral cavity and, to a lesser extent, the concentration of salivary antimicrobials. Eventually, changes in the saliva affect the resident bacterial flora concentrations and composition, promoting the proliferation of yeasts. Other contributing factors affecting mucosal homeostasis are tobacco use, which increases the risk of fungal infection during cancer therapy, the use of antibiotics and steroids, the presence of oral prosthesis, and poor oral hygiene [13]. During RT for HN cancers, fungal infections often feature *C. albicans*, and this condition favors the onset of other *Candida* infections, such as *C. tropicalis, C. glabrata*, and *C. krusei* [26].

The main clinical lesions resulting from fungal infection of the oral cavity may involve any part of the oral cavity or pharynx [13]. New advances in RT such as intensity-modulated RT facilitate very precise treatment with high-dose coverage of tumors, while avoiding the irradiation of adjacent structures (more specifically, salivary glands). These new techniques decrease the frequency and intensity of hypo-salivation, reducing the need for dysbiosis-based discontinuation of cancer therapy. Importantly, intensity-modulated RT may reduce the impact of high-dose RT on fungal virulence profile, which was noticed in *C. tropicalis* [25].

It is important to underline that other kinds of infections have been described in patients undergoing RT and/or CT. An Italian study showed that patients hospitalized in the Head and Neck Medical Oncology Unit were prone to oral cavity bacterial infections, with a predominance of Gram-negative infections that can lead to pneumonias [27]. Infections may also be localized at the site of surgery, evolve into respiratory infections, or become systemic [27]. During OM, a dramatic increase in Lactobacilli and *Staphylococcus aureus* have been observed that was not linked to upcoming infections but may explain the occurrence of dysbiosis [28].

## 5. RT-Dependent Oral Infections: What We Know about Oral Anti-Fungal Immunity

As it has been extensively demonstrated [13], the human commensal *C. albicans* represents one of the main microbial risks during OM, and this yeast is able to promote opportunistic infections in the immunocompromised host undergoing RT. To understand how to resist *Candida* dissemination following barrier leakage, as occurs with RT, several studies have focused on molecularly dissecting the innate and adaptive host mechanisms that maintain host–fungal symbiosis in normal conditions.

Among the innate defenses, the oral microbiota has been shown to have important roles in counteracting fungal infection in recent studies. Other effective innate defenses in the oral cavity are: secretion of mucins, antimicrobial peptides, and the release of salivary proteins and secretory IgA [29]. Using a mouse model of oropharyngeal candidiasis, Millet et al. showed the important role of oral saliva IgA in regulating *Candida* infections and oral dysbiosis [30]. In this regard, pro-inflammatory cytokines in the saliva, such as IL-6 and IL-1β, are reportedly produced at high levels upon RT and during the development of OM in patients with HN cancers [31]. These findings suggest that, to prevent the onset of mucosal candidiasis in the oral cavity, it would be more efficacious to reconstitute the epithelial layers than to induce strong inflammation although antimicrobials. Indeed, it has been shown that the production of epidermal growth factor during OM correlates with a low grade of clinical OM. The degree of basal epithelial cell proliferation reduces the susceptibility of the oral mucosa to OM. Thus, the reduction in epidermal growth factor post-radiation suggests that epithelial layer regeneration may prevent mucosal damage, promote healing, and, eventually, avoid fungal infection [31,32].

In this context, the role played by the microbiome changes in modulating tissue regeneration and oral immunity is pivotal. It is interesting to know that different studies highlight an increase of *Lactobacilli* spp. in the same timeframe of *Candida* oral infection in RT OM [11,28,33]. This increase was not linked to upcoming infections, but it may explain that in dysbiosis, probiotic bacteria may be involved in regulating tissue healing and immunity. Also, anti-bacterial treatment strategies for OM have been basically unsuccessful [34].

Regarding adaptive immune cells important to regulate anti-fungal responses in the oral cavity, RNA sequencing of murine tongues with *Candida* oral infections was recently performed to determine the transcriptional host response. During *Candida* infection, the tissue displayed the upregulation of adaptive host responses, such as the immune network related to IgA production. The specific binding of IgA to the fungus prevented fungal colonization and spread, resulting in reduced oral inflammation. Moreover, immunofluorescence microscopy showed that plasma cells were located in the tongue, where *Candida* was able to create foci of colonization [30].

In addition, T lymphocytes were also investigated to describe the role of adaptive immunity against fungal infection in the oral mucosa. A delicate balance between Treg cells, particularly those present in the oral cavity at the tongue epithelial regions, and Th17 cells also regulates oral inflammation during infection. Although IL-17 immunity is strongly induced, it seems that the stimulation of both cell types regulates fungal oral clearance [35]. These results explain how antifungal oral immune responses are effective against the fungus when balanced between anti-inflammatory cells (Tregs) and inflammatory cells (Th17).

Possible interactions between T follicular cells and B cells are also underestimated in this context, and preclinical studies may better delineate the role of cell-mediated immunity and the secretion of protective IgA in establishing infection after RT. In contrast, in the last few years, both the innate and adaptive roles of IL-17 have been under thorough investigation [36]. Through observations of preclinical models, IL-17 has been seen to activate early and late pro-inflammatory responses against invading pathogens during oropharyngeal candidiasis [36]. IL-17 induces antifungal oral responses, such as the release of antimicrobial peptides or defensins. For OM-dependent *Candida* infections, it is vital that we understand whether inflammation is the main cause of dysbiosis and the importance of the IL-17 immune signature in resistance to *Candida* infection.

To underline the importance of understanding IL-17 immunity in fungal infection in OM, Aggor et al. [37] demonstrated that, during oral *Candida* infection in a murine experimental model, basal epithelial cells expressed high levels of IL-22 receptor (R), which binds IL-22 also released by Th17 cells. The exposure of mouse tongues to IL-22 induced the proliferation and survival of basal epithelial cells, suggesting that IL-22 has a role in regenerating the oral mucosa. There is particular interest in IL-22, as it has been widely demonstrated to control fungal infection in the gut and in vaginal tissue [38]. Therefore, future studies are required to clarify whether, during RT and the development of OM, there is a reduction in IL-22 levels as well as the expression of the IL-22 R by epithelial cells. Considering the importance of IL-22 in maintaining mucosal barrier integrity and shaping the microbial community, this cytokine may also play an interesting protective role in the oral cavity post-RT. Indeed, the hallmark of candidiasis in autoimmune polyendocrinopathy-candidiasis-ectodermal dystrophy is the production of highly neutralizing autoantibodies against type I interferons and IL-22 [39].

Thus, oral immunity (innate or adaptive) modulation (regulation of Th17 cells, maintenance of the Treg balance, IL-22 induction, IgA production) may be greatly important to prevent or to treat fungal infections post-RT.

## 6. Prevention and Treatment of OM and Related Infections

OM can be prevented and treated in a number of ways, including oral hygiene regimes, analgesia, and drugs to combat dry mouth, either applied locally (Table 1) or systemically (Table 2). In the tables, the treatments are listed from most to less commonly used.

The idea that oral dysbiosis may contribute to OM has also been suggested by preclinical studies: evidence shows that the total number of bacteria will potently increase as long as the oral epithelium is enriched in ulcerations. The hypothesis is that bacterial cell wall products or pathogen-associated molecular patterns activate innate cells surrounding the oral lesions, promoting local tissue cytokine storms [12]. Thus, oral hygiene is one of the most effective and frequent approaches in reducing the risk of OM, although those treatments may be detrimental in facilitating the reduction of commensal microbes beneficial in OM.

The oral cavity contains many Gram-negative bacilli, which may have a role in mucositis development; hence, the concept of “decontamination” has been developed to minimize microbial colonization and reduce the risk of OM. Furthermore, decontamination may reduce the grade of OM, as the microbial colonization of lesions and even oral dysbiosis have been hypothesized to exacerbate OM severity [40,41]. On the other hand, part of the oral microbiota may be essential for mucosa recovery.

The Multinational Association of Supportive Care in Cancer (MASCC) [22] and the International Society of Oral Oncology (ISOO) [42] guidelines recommend the use of a standardized oral care protocol. This includes oral rinsing with a non-irritating solution such as saline solution or sodium bicarbonate to increase the quality of the saliva, daily ultra-soft tooth brushing with a fluoride toothpaste, maintaining a low-sugar and non-acidic food and drink diet, avoiding smoking and alcohol, and refraining from flossing when the platelet count is low because of the risk of bleeding. Furthermore, pre-existing oral pathology, e.g., dental caries, periodontal lesions, pulpal disease, and oral xerostomia, are linked to increased bacterial colonization and severe OM [7], and these should be treated before initiating RT.

Other approaches include the administration of keratinocyte growth factor, an epithelial mitogen that reduces the levels of reactive oxygen species by activating nuclear factor (erythroid-derived 2)-like 2 [42,43]. Amifostine, a free-radical scavenger, antioxidant, and cytoprotective agent, is administered subcutaneously or by venous infusion before RT to reduce the incidence of moderate to severe xerostomia [44,45,46,47,48], and blood pressure must be monitored before, during, and after the infusion. Additionally, low-energy helium–neon laser application before RT is associated with a significant reduction in the duration and severity of OM [49].

Nutritional support, pain control, prophylaxis, and/or the treatment of secondary infections are the main cornerstones of OM management [44,50].

## 7. Can We Harness the Microbiota to Counteract RT-Dependent Fungal Infections?

### 7.1. The Oral Microbiota

The benefits of the oral microbiota, which represents one of the major immune barriers counteracting fungal infections, have been underestimated for many years. Nowadays, thanks to the recent advances in DNA sequencing and shotgun metagenomics, the oral microbiota is taken more and more into account in the context of OM and fungal infections occurring in OM [71].

The human oral microbiota is considered to be the second most intricate microbiota in the body, second only to that of the intestine. Recently, the Human Oral Microbiome Database (HOMD) has been made available to researchers providing a better understanding of the distribution of microbial taxa in the oral mucosal layers and in various oral diseases in humans [72]. One of the greatest limitations is that a considerable percentage of oral microorganisms is unculturable, as much as 20% to 60%. The oral microbiome is dominated by five major phyla, Firmicutes, Bacteroidetes, Proteobacteria, Actinobacteria, and Spirochaetes, which comprise 94% of the taxa detected. An imbalance in the oral microbial flora (dysbiosis) contributes to oral diseases, such as dental caries and periodontitis, secondary infections, oral mucosal diseases, and systemic diseases, such as gastrointestinal and nervous systemic diseases [73].

Numerous studies have investigated the composition of the human oral microbiome and changes in the different oral niches in different conditions, varying from oral tumors to periodontitis; for example, compared with healthy individuals, diabetics presented *Fusobacterium, Peptostreptococcus*, *Gemella, Streptococcus, Leptotrichia, Filifactor, Veillonella, Terrahemophilus*, and elevated levels of *Capnocytophaga*, *Pseudomonas, Bergeyella, Sphingomonas, Corynebacterium*, *Propionibacterium*, and TM7 [74]. Sequencing revealed elevated numbers of *S. mutans* and *Lactobacillus salivarius* in the mouths of smokers compared to non-smokers [75]. The differences in microbiome composition associated with caries have also been described [76], and healthy controls presented more *Bacilli* and *Gammaproteobacteria*, *Rothia*, and *Aggregatibacter,* while diseased patients presented more *Clostridiales* and *Bacteroidetes* [76].

Pivotally, high-throughput pyrosequencing of 16S DNA also revealed changes in the microbial composition of oral plaques during HN RT. Further, they found a negative correlation between the number of operational taxonomic units (OTUs) and radiation dose, although they identified a microbiome core that was unaffected by RT treatment [9,77]. In saliva, the core microbiome was similar to the oral plaque microbiome [78]. In both studies, at the genus level, RT induced strong fluctuations in *Streptococcus* numbers and a reduction of OTUs [9]. A striking increase in *Actinomyces*, *Veillonella*, and *Rothia* was also observed when the RT dosage was increased [77]. Moreover, there is evidence that RT may also affect bacterial virulence and metabolism [79].

OM severity was assessed and scored in a study of patients with nasopharyngeal carcinoma undergoing RT, and 16S rRNA gene sequencing and analysis of the retropharyngeal mucosa was used to investigate dynamic changes in the oropharyngeal bacterial profile. The data revealed an association between the bacterial profiles with the severity of mucositis. In particular, genera such as *Phenylobacterium*, *Acinetobacter*, *Burkholderia*, *Sphingomonas*, *Azospirillum*, *Rhizobium*, *Hydrogenophaga*, *Paracoccus*, and *Nocardioides* were positively associated with an increase in OM severity [20]. It is still very difficult to ascertain whether the variations in oral mucosal tissues due to RT induce local changes in the core microbiome or vice versa [34]. More studies are needed to better delineate the causality of the phenomenon during OM. Similarly, a better understanding of the possible association between the microbiome and the occurrence of fungal infections is needed [34]. Thus, an oral microbial signature may accompany the variations in mucosal homeostasis that predispose patients to candidiasis and, therefore, could help in preventing the onset of infection, particularly those infections that are difficult to eradicate once established.

Recently, the posterior pharyngeal mucosa was sampled to understand variations in the oral microbiota upon 0, 20, 30, 40, 50, 60, and 70 Gy RT sessions [10]. The overall microbial signature of oral mucosal tissue was found to change significantly during radiation: *Pseudomonas*, *Treponema*, and *Granulicatella*, were significantly and positively associated with RT dose, whereas bacteria such as *Prevotella*, *Fusobacterium*, *Leptotrichia*, *Campylobacter*, *Peptostreptococcus*, and *Atopobium* were negatively associated [80]. The relative abundance of *Fusobacterium*, *Treponema*, *Porphyromonas*, and *Prevotella* decreased radically and was significantly and stably lower after the onset of OM. The authors suggested that this signature may play a detrimental role in exacerbating inflammation in the oral tissue. This particular combination of bacteria is often reported in the oral cavity and in different pathogenic conditions. For instance, a similar signature was associated with *Candida* infections in patients undergoing stem cell transplantation [81].

Therefore, descriptions of microbial shifts during RT may be fundamental to our apprehension of how the bacterial/yeast community is able to symbiotically interact with the mammalian host under physiological conditions and if there are any “good” commensal bacteria combinations that could help to control the fungal spread that coincides with tissue leakage. This concept, which is still very empirical, is the basis of the administration of probiotics to counteract gastrointestinal, vaginal, or skin dysbiosis. With modern metagenomics techniques, it may be possible to rationalize why therapy with probiotics has shown such efficacy in the treatment of human diseases.

### 7.2. Novel Insights into OM Therapy Based on Microbiome Modulation

The use of probiotics to treat OM is still fairly uncommon because of several problems with their administration and efficacy. To be effective, oral probiotics should adhere to the oral tissues and form a biofilm together with the innate microbial components, and they, ideally, should not contain fermentable sugars that induce caries. Moreover, it is unclear how pharmaceutical probiotic formulations should be prepared to prevent them from being swallowed, which would reduce the efficacy or induce undesirable systemic effects. In this respect, *Lactobacillus brevis* CD2 was used to prevent OM in multicentric randomized studies involving patients with HN cancers [17,82]. CD2 produces high levels of arginine deaminase and, thus, reduces arginine availability in the oral cavity. The arginine-dependent growth of microorganisms implicated in inflammatory processes is consequently reduced. Moreover, LB CD2 lowers the amount of arginine available to arginases, resulting in less nitrous oxide production and the subsequent attenuation of inflammatory processes. However, the studies obtained contrasting results, highlighting the need for a better understanding of how to apply probiotics effectively.

In addition, metatranscriptomics applied to metagenomics may help to predict how dysbiosis occurs and how the microbiota transcriptionally shifts during dysbiosis. Information related to the transcriptional control of oral commensals, or of probiotics in general, during dysbiosis could lead to the discovery of novel immunomodulatory molecules, also known as postbiotics [83,84]. Several classes of postbiotics have already been described ranging from metabolites to cell wall components. Recent advances in metatranscriptomics also revealed how different biosynthetic gene clusters coding for biochemical and secondary metabolism pathways were activated at host tissue niches by the microbiome [85].

We have demonstrated that the probiotic *Lactobacillus reuteri* [38], which is particularly prominent in the gastric mucosa and small intestine, is able to degrade the amino acid tryptophan into indole-derivatives with postbiotic activity. The indoles can bind to the xenobiotic receptor aryl hydrocarbon receptor (AhR), and the activation of AhR, particularly of innate lymphocytes in the small intestine, stimulates an IL-22-dependent response. The release of IL-22 is effective against the fungus *Candida,* in reducing intestinal leakage, and, as discussed above, this cytokine is active on epithelial barriers with elevated expression of its receptor, IL-22 R. The activation of AhR also leads to anti-inflammatory effects mediated by the expansion of Tregs and production of IL-10 [38]. In a mouse model of chemotherapy-induced oral mucositis, *Lactobacillus reuteri* DSM 17,938 and ATCC PTA 5289 ameliorated oral inflammation by increasing the epithelial thickness and reducing oxidative stress via nuclear factor (erythroid-derived 2)-like 2 [86]. The postbiotic activity of *Lactobacillus paracasei* 28.4 against *Candida auris* was also recently reported, demonstrating the importance of postbiotics in directing antimicrobial activities and regulating biofilm formation [83]. Therefore, multiple lines of evidence suggest that both microbial changes and the immunomodulant properties of certain probiotic elements may be exploited for OM therapy and/or to control fungal infections in OM.

## 8. Future Directions

In recent years, technological advances in three-dimensional (3D) culture have revealed the benefits of employing human tissue models to better understand the basic mechanisms related to tissue inflammation, such as the processes underway in OM.

Studies in which tongue-derived adult stem cells have been used for tongue organoid cultures have led researchers to rethink the OM models [87]. A shift towards the use of 3D human organoids is already evident in investigations of mucositis in the gut, as has been seen with the exploitation of “tumoroids” in the study of tumors [88]. A substantial limitation of these models is the absence of human immune cells and vasculature, but the cocultivation of heterogenous organoids to better study system complexities is already under development. Organoids derived from tongues may provide valuable alternatives to murine models when studying changes in the epithelial compartments in response to RT. Moreover, the addition of microbial infections to the organoids to trigger biofilm formation or antifungal innate responses is also possible. Thus, the development of 3D cultures is pivotal to the in vitro creation of a human/microbe ecosystem for future discoveries.

As human microbiota technology in sequencing/identification/functionality becomes increasingly feasible and available for clinicians, researchers will be able to better define the microbial signatures associated with pathological conditions. Therefore, in addition to metagenomics, it is important to identify supporting metatranscriptomics that will clarify the transcriptional activity of microbes in human tissues.

These advances will also help to identify the function and the meaning of certain fluctuations of the core microbiome, such as the demonstrated increase of Lactobacilli in the occurrence of oral *Candida* infections during OM [28].

## 9. Conclusions

The oral cavity is one of the most biologically complex environments in the human body, and the roles of resident microflora in OM development in patients receiving RT are still being debated. The recent studies described here are encouraging the scientific community to reconsider how nonpathogenic oral microbes can be used to combat infection and/or inflammatory conditions. On the other hand, many bacteria are most likely implicated in the pathogenesis and maintenance of inflammatory diseases of the oral cavity and most of them may favor tissue epithelial leakage, thus leading to opportunistic fungal infections.

This makes it clearer that future development into the discovery of the molecular interactions and biological functions of the core microbiome with host tissue will better clarify the cause–effect of dysbiosis in OM and eventually in counteracting opportunistic fungal infections.

Furthermore, researchers have developed human mini-3D oral organoids and already exposed them to RT to examine its multiple side effects on the mucosa [88]. Therefore, exposing oral organoids to oral microbes could provide a novel way to identify mechanisms of host–commensal interactions and eventually to discover novel microbiome-based therapy aimed at solving the fastidious host-microbiome breakdown of symbiosis in RT induced OM.

## Figures and Tables

**Figure 1 jof-07-00290-f001:**
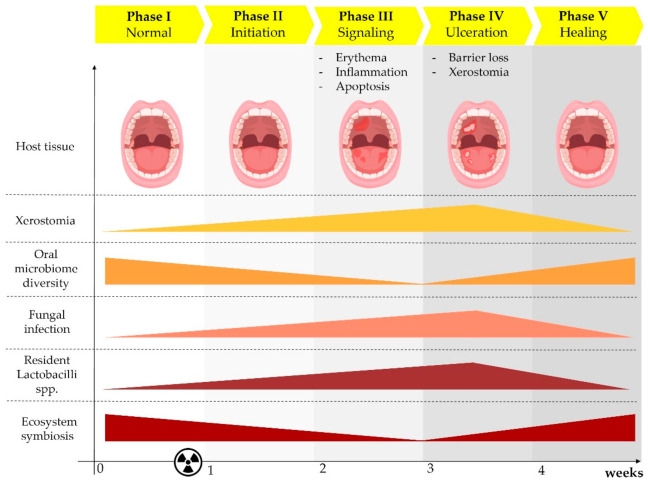
Radiation therapy (RT) induced multiple side effects during the oral mucositis (OM) stages. The clinical effects of RT in the host oral mucosa are highlighted in the upper panel (host tissue). Below, oral microbiome diversity is associated with reduced ecosystem symbiosis (dysbiosis) and the increased fungal infection occurrence. In parallel, the increase of *Lactobacilli* sp. is significantly high compared to healthy controls, when patients experience xerostomia. Novel studies reviewed here highlight the importance of more deep investigations to better explain the causality of these significant fluctuations for a more rational development of anti-inflammatory or antifungal therapies in OM.

**Table 1 jof-07-00290-t001:** Locally applied agents for the treatment of OM listed from most to less commonly used.

Agent	Effect	Reference
**Glycyrrhetinic acid/povidone/sodium hyaluronate gel**	Adherence to the mouth mucosal surface, soothing oral lesions	[51]
**L-glutamine**	Counteraction of RT-induced metabolic deficiencies	[52]
**Local anesthetics**	Short-term relief of OM-associated pain (e.g., diphenhydramine, viscous xylocaine, lidocaine, and dyclonine hydrochloride)	[53]
**Artificial saliva sprays**	Alleviate mucosal dryness in mild cases of OM	[54]
**Vitamin E (tocopherol)**	Reduction of oral mucosa oxidative damage and, consequently, incidence of symptomatic OM	[55,56]
**Sodium alginate**	Reduction of OM-linked discomfort and OM severity	[57]
**Zinc-L-carnosine**	Physical barrier protection and repair of damaged areas	[58]
**Polideoxyribonucleotide (PDRN)**	Regenerative and anti-inflammatory device	[53,59,60]

**Table 2 jof-07-00290-t002:** Systematically applied agents for the treatment of OM listed from most to less commonly used.

Agent	Effect	Reference
**Cyclooxygenase-2 inhibitors**	Suppression of NF-κB, reduction of pro-inflammatory cytokine production, inhibition of angiogenesis	[61,62]
**N-acetylcysteine**	Antioxidant agent that suppresses NF-κB activation	[63]
**Minor Analgesics and Opioids**	Mitigation of OM-related pain	[44,64]
**Azelastine**	Potent second-generation selective histamine antagonist used as an anti-inflammatory and antioxidant agent	[65]
**Systemic corticosteroids**	Anti-inflammatory and antioxidant agent	[66]
**Antibacterial agents**	Prophylaxis of aerobic (e.g., *Pseudomonas* spp. and *Staphylococcus* epidermidis) and anaerobic (e.g., Bacteroides spp. and Veillonella spp.) bacterial infections	[67,68,69]
**Systemic administration of Fluconazole**	Prophylaxis of fungal infections, which can complicate the clinical scenario, especially in immunocompromised patients. Fluconazole significantly reduced the severity of OM and the risk of RT interruption	[70]

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
