# Peer review of "Breakdown of Symbiosis in Radiation-Induced Oral Mucositis"

_jof, 2021, doi:10.3390/jof7040290_

Round 1

Reviewer 1 Report

The authors have addressed most of my original concerns. The English language and style have improved, but there remains a few issues that should be corrected. Please avoid the use of common language including, but not limited to, "absolutely of interest"; line 210. Since new sections have been added, they should be evaluated for proper grammar as well. 

Author Response

We thank the reviewer for the suggestions. We have provided a new version of the manuscript where changes are highlighted in yellow.  We have revised the language as suggeted.

Reviewer 2 Report

The Authors have addressed my comments and the manuscript has significantly improved. 

An interesting association is the increase in lactobacilli in the oral microbiome with grade3-4 and also increase candida infections at the same time. Especially since Lactobacilli are normaly associated with protection against Candida infections as the authors also describe with probiotic L. reuteri treatment

I would suggest the authors to carefully revise their text another time as there are still some typo's and sentences with errors and it would be a shame if these would remain in the final manuscript.

The section on OM grading is quite technical and could benefit from some revisions to make it more accessible to a broader audience

Author Response

We thank again the reviewer and we have highlighted in yellow the changes in the text as suggested.

This manuscript is a resubmission of an earlier submission. The following is a list of the peer review reports and author responses from that submission.

Round 1

Reviewer 1 Report

Summary

The review by Ingrosso and colleagues describes oral mucositis, a common acute complication of radiotherapy. During this disease the symbiosis of the oral microbiota is disturbed, which greatly contributes to the pathogenesis and persistence of oral mucositis. As with many disease associated with a disturbed microbiota the use of probiotics and therapeutic microorganisms is explored as a novel treatment strategy.

Although the review is quite comprehensively describing oral mucositis, only when coming past the middle of the review it becomes clear why this was submitted to a microbiological (mycology) journal. This is a review for the Journal of Fungi, but the introduction about the fungi in question, the Candida species, is almost nonexistent. It is not explained how it can affect the patient, how severe it is, and the pathogenicity mechanisms.

Given the eventual focus in the end of the review and the outlook, the manuscript would benefit from substantial restructuring of the focus. Otherwise it is impossible for the reader to understand the importance of probiotics or anti-fungal immunity. In this same context, it would be useful to explain what is known about the interactions between healthy microbiota and Candida. Especially since the title states “breakdown of symbiosis”, this needs to be discussed in more detail. The symbiosis between Candida and microbiota is not emphasized enough. It would be important to mention early on (section 1) that the balance and dysbalance of the microbiota plays a key role in oral mucositis. Also in the following sections on preventive measures and treatment the connection with the microbiota/opportunistic pathogens such as Candida needs to be made more clearly. Such changes would be essential to increase the interest of the readership of Journal of Fungi in this review.

The manuscript needs to be polished, as there are quite some grammar mistakes, spelling, and formatting inconsistencies.

Below are my specific comments

Major comments:

  1. The abstract is slightly misleading. At first read I was under the impression that this was an original article. Please phrase clearly that this is a review summarizing the literature on OM, with a focus on how the imbalance in the microbiota is associated with opportunistic infections and how probiotics can be used for treatment.

  1. At the end of the first section (Lines 39-44) clinical signs and consequences are mentioned together. Please rewrite in two sentences to separate the clinical signs and consequences.

In this respect, there is quite some literature on how radiotherapy has great consequences on the oral microbiota. It would be great if this could already be mentioned here to change the scope to more microbial focused.

  1. Section 2 on grading would benefit from an introductory sentence explaining why grading is necessary or important. Further, there are no references in this paragraph.

Regarding shifting to a microbial oriented scope; is there any association between the grade of oral mucositis and changes in the oral microbiome?

  1. In section 3 there are no references cited in the first few sentences. Further, there are some instances of repetition. In lines 58-60 several risk factors are listed. Then in sentences 61-63 several of these risk factors are listed again. Regarding shifting to a microbial oriented scope; is there any information on carriage of specific members of the microbiota being risk factors for oral mucositis???

  1. The fourth section is on preventive measures, but also includes various treatments. Please adapt the title.

In line 70 Remove “Treatment of OM are reported in Table 1.” as it is discussed below. In this way first prevention and then treatment is discussed.

Further the preventive measures for OM are listed with bullet points, and in this way it reads almost like patient information rather than a review. Please restructure this section. Also consider what is crucial for the understanding of the rest of the review and what can be rather left out or summarized.

On line 84 it is mentioned that Keratinocyte growth factor reduces reactive oxygen species. Mention the connection between ROS and OM manifestations/ symptoms. Change “nuclear factor (erythroid derived 2)-like 2:[15, 16].” To: nuclear factor (erythroid-derived 2)-like 2 [15, 16]. Also provide the abbreviation NRF2 since this is again later mentioned in the text.

Line 66 is also the first time the authors mention microbes. Here it is mentioned as if it was clear to the reader that microbes play an important role in OM. But from the first 3 sections this is unclear. Given the scope of the journal, and the rest of the review, it would be highly beneficial to already mention in the first sections that the oral microbiota plays a key role in the pathogenesis of OM; see suggestions above.

In this respect, section 4 can also consequently be adapted in this respect. Most of the preventive measures listed (Smoking, Alcohol, Denture use, low sugar and non acidic food, oral rinsing with saline) will significantly impact the oral microbiota. But are currently simply listed without any context to why these preventive measures are important in shaping the microbiota. With respect to the previous comment, rewriting this more in the scope of the review will make the section look less like “patient information”

Further, Line 66 mentions microbial overlap. What is specifically mentioned by this?

  1. Tables 1 and2: need to be reformatted, because in some cases it is difficult to distinguish which effect belongs to each agent.

lines 93-95 referring the tables should be combined to: Locally and systemically applied agents to treat OM are listed in Table 1 and Table 2 respectively.

That the therapies are listed from the most to the less commonly used should be mentioned in the table caption. For example: Table 1 title: Locally applied agents for the treatment of OM listed from most to less commonly used. Same for Table 2.

  1. In section 5 Line 106 “mucosal barrier impairment and radiation-related dysbiosis favors the colonization and invasion of commensal bacteria and fungi.” Commensal microbes are by definition already colonizers of the mucosal surfaces, it should rather be stated that these overgrow and that the dysbalance of allows these normally harmless commensals to cause disease.

In lines 111-113 it is mentioned that Candida species are the main commensal yeast in the GI-tract. I would say that this information is redundant within the scope of this review.

In line 114 the citation is missing

In lines 127-129 the information on how tobacco, antibiotics, oral prosthesis and poor oral hygiene contributes should be included in the section on preventive measures for oral mucositis.

Lines 131-133 the clinical lesions due to fungal infection either need to be more clearly explained or their names not mentioned at all if they are not relevant. Knowing only the names of the lesions doesn’t add extra information.

  1. To better organize the flow of the review the oral microbiota section should be placed before the opportunistic fungal infection section. Furthermore, another example that the microbiota needs to be integrated in the first few paragraphs is that prophylaxis treatment is already mentioned in section 4 and is mentioned in the fungal infection section that dysbiosis leads to fungal infection. Such redundant overlaps (E.g. in line 215 microbial dysbiosis is explained but this explanation is already discussed above) and duplications can be avoided by thoroughly restructuring the review.

  1. Lines 217-220 consider whether the information about the eHOMD is relevant for the topic. If so, move information to line 208 where the database is discussed. Or only discuss the most recent database, which is most relevant.

  1. Figure 1b needs to be elaborated. Not a lot of information on the tissue parameters is given. For example, how is budding studied in these models?

The manuscript needs to be polished, as there are quite some grammar mistakes, spelling, and formatting inconsistencies. Se below my specific suggestions:

  • Make sure consistently a space before the references is inserted
  • Line 10 remove “highly” before “predispose”
  • Line 14 remove “multiple” before “benefits”
  • Line 16 remove “murine organoids” as only human organoids is discussed in this review
  • Line 16-17 change order to: have already been exposed
  • Line 19 change “host commensals” to: host-commensal
  • Line 34 a new paragraph is started, but reading the text this looks unneccesary that shouldn’t be a new paragraph. Further rather place reference [7] at the end of the sentence or after “erythema”.
  • Line 35 change “the first signs of OM (Figure 1).” To: the first visible sign of OM (Figure 1).
  • Line 35-38 Consider rewriting the sentence “In the subsequent epithelial phase the damage of the membrane basement with loss of the protective barrier determines different degrees of desquamation and ulceration, which appears when the cumulative dose reaches 30 Gy [80].” To: Damage of the basement membrane beneath the epithelial cell layer leads to loss of the protective barrier which determines different degrees of desquamation and ulceration when the cumulative dose reaches 30 Gy.
  • Line 41 Change “narcotic analgesics, prolonged hospitalization are” to: narcotic analgesics and prolonged hospitalization are
  • Line 42 change “led” to: lead
  • Line 48 change “do not” to: does not
  • Line 51-52 change “appear.” to: appears
  • Line 53-54 change “it is wide spreading used.” to: it is widely used.
  • Line 60 change “oral hygiene, dental status” to: oral hygiene and dental status
  • Line 68 First time the serial comma is used before “and”
  • Line 68-69 change “xerostomia are linked with an increased bacterial colonization and severe OM [7]; consequently it is recommended their cure before the start of RT.” to xerostomia are linked with increased bacterial colonization and severe OM [7]; consequently their cure is recommended before the start of RT.”
  • Line 86 Add by before “venous infusion”
  • Line 89 Change “applied” to: application
  • Line 92 Change “main cornerstones in the” to: main cornerstones of
  • Line 99 Write: Opportunistic fungal infections instead of “fungal opportunistic infections”
  • Line 113 replace “of” by: for
  • Line 115-116 Radiotherapy could be abbreviated RT.
  • Lines 135-139: consider rewriting. This is a very long sentence that could be split in multiple.
  • Line 142 write: main microbial risks instead of: “main microbial risk”
  • Line 144 no need to split in two paragraphs.
  • Line 144 write: to resist Candida dissemination instead of: “to resist to Candida dissemination”
  • Line 148-149 Change “important and essential niche” to: an essential niche
  • Lines 149-150 consider writing “In addition, the innate defenses with the highest importance in the oral cavity are basically related to…”
  • Line 153 remove interestingly.
  • Line 162 change “regulates” to: regulate
  • Line 167 correct “contest” to: context
  • Line 170 insert: in between “IL-17” and “both”
  • Line 172 write: early and late proinflammatory response instead of “proinflammatory early and late response”
  • Line 175 correct “dependend” to: dependent
  • Lines 175-177 rewrite, because it says “it will be important to understand ... IL-17 immune signature importance”.
  • Line 177 write: to resist Candida infection instead of “to resist to Candida infection”
  • Line 178 keep spelling of “proinflammatory” consistent
  • Line 180 change “may suggest” to suggests
  • Line 181-182 Change sentence to read: it would be more essential to reconstitute the epithelial layers than to induce strong inflammation although it is antimicrobial
  • Line 184 change the verb “influence” for one more specific. (e.g. epithelial proliferation increases/decreases the sucseptibility.
  • Line 189 indicate that IL-22R is the IL-22 receptor
  • Keep spelling of IL-22R consistent line 189 vs line 195
  • Correct placing of ref [52], [54], [55].
  • Line 206 correct “contest” to: context
  • Line 206 change “in to” to: into (also correct this in other parts of the manuscript)
  • Line 210 write limitations instead of “limitation”
  • Lines 210-212 consider rewriting so as not to repeat information. Also use: culturable and unculturable instead of “cultivable” and “uncultivable”
  • Line 212 correct spelling of “phila” to: phyla
  • Line 220 remove the comma after “researchers”
  • Line 226 insert: and before “TM7”
  • Line 227 change “non smokers” to: non-smokers
  • Line 230 remove the capitalization of the P in “Gammaproteobacteria”
  • Line 235 change “was not dissimilar from” to: was similar to
  • Line 237 change reference formatting to: [61, 62]
  • Line 247 insert:that before “positively correlated”
  • Line 249 change “viceversa” to: vice versa and italicize
  • Line 258 a new paragraph is not needed here and the word “Here” needs to be removed at the end of the sentence
  • Line 261-262 “and” is italicized.
  • Line 261-262 Remove etc., this does not provide any useful information
  • Line 263 consider not starting a new paragraph.
  • Line 270 consider changing “knowledge” for another alternative such as understanding.
  • Line 272 change the expression “good combination” for something more specific, such as: beneficial combination.
  • Line 281 write: fermenting instead of “ferment”.
  • Line 282 repleace “very puzzling” by: complicated
  • Line 292 change reference formatting to: [70, 71]
  • Line 299 correct “in to” to: to
  • Line 315 remove “The”
  • Line 320 change “organoid” to: organoids
  • Line 323 change “it is already under development the possibility” to: it is already under development with the possibility
  • Lines 321-324 consider rewriting for more clarity and correct word order in the sentence.
  • Lines 326-327 correct to: Microbes can also form biofilms or trigger antifungal innate immune responses

Author Response

1
Review# 2
Summary
The review by Ingrosso and colleagues describes oral mucositis, a common acute complication of radiotherapy. During this disease the symbiosis of the oral microbiota is disturbed, which greatly contributes to the pathogenesis and persistence of oral mucositis. As with many disease associated with a disturbed microbiota the use of probiotics and therapeutic microorganisms is explored as a novel treatment strategy.
Although the review is quite comprehensively describing oral mucositis, only when coming past the middle of the review it becomes clear why this was submitted to a microbiological (mycology) journal. This is a review for the Journal of Fungi, but the introduction about the fungi in question, the Candida species, is almost nonexistent. It is not explained how it can affect the patient, how severe it is, and the pathogenicity mechanisms.
Given the eventual focus in the end of the review and the outlook, the manuscript would benefit from substantial restructuring of the focus. Otherwise it is impossible for the reader to understand the importance of probiotics or anti-fungal immunity. In this same context, it would be useful to explain what is known about the interactions between healthy microbiota and Candida. Especially since the title states “breakdown of symbiosis”, this needs to be discussed in more detail. The symbiosis between Candida and microbiota is not emphasized enough. It would be important to mention early on (section 1) that the balance and dysbalance of the microbiota plays a key role in oral mucositis. Also in the following sections on preventive measures and treatment the connection with the microbiota/opportunistic pathogens such as Candida needs to be made more clearly. Such changes would be essential to increase the interest of the readership of Journal of Fungi in this review.
We thank the reviewer for the suggestions given. We tried to make changes in order to better describe the main focus of the manuscript.
The manuscript needs to be polished, as there are quite some grammar mistakes, spelling, and formatting inconsistencies.
We have reviewed the entire manuscript as suggested
Below are my specific comments
Major comments:
1. The abstract is slightly misleading. At first read I was under the impression that this was an original article. Please phrase clearly that this is a review summarizing the literature on OM, with a focus on how the imbalance in the microbiota is associated with opportunistic infections and how probiotics can be used for treatment.
We agree that the abstract originally was misleading, now we have added more details to better describe the review article.
2. At the end of the first section (Lines 39-44) clinical signs and consequences are mentioned together. Please rewrite in two sentences to separate the clinical signs and consequences. In this respect, there is quite some literature on how radiotherapy has great consequences on the oral microbiota. It would be great if this could already be mentioned here to change the scope to more microbial focused.
2
We amended the text as suggested (lines 24-68).
3. Section 2 on grading would benefit from an introductory sentence explaining why grading is necessary or important. Further, there are no references in this paragraph. Regarding shifting to a microbial oriented scope; is there any association between the grade of oral mucositis and changes in the oral microbiome?
We amended the text as suggested (lines 70-111).
4. In section 3 there are no references cited in the first few sentences. Further, there are some instances of repetition. In lines 58-60 several risk factors are listed. Then in sentences 61-63 several of these risk factors are listed again. Regarding shifting to a microbial oriented scope; is there any information on carriage of specific members of the microbiota being risk factors for oral mucositis???
We amended the text as suggested (lines 113-136).
5. The fourth section is on preventive measures, but also includes various treatments. Please adapt the title.
We changed the title as suggested (line 298).
6. In line 70 Remove “Treatment of OM are reported in Table 1.” as it is discussed below. In this way first prevention and then treatment is discussed.
We edited the text as suggested (lines 299-301)
7. Further, the preventive measures for OM are listed with bullet points, and in this way it reads almost like patient information rather than a review. Please restructure this section. Also, consider what is crucial for the understanding of the rest of the review and what can be rather left out or summarized.
We edited the text as suggested (lines 299-337).
8. On line 84 it is mentioned that Keratinocyte growth factor reduces reactive oxygen species. Mention the connection between ROS and OM manifestations/ symptoms. Change “nuclear factor (erythroid derived 2)-like 2:[15, 16].” To: nuclear factor (erythroid-derived 2)-like 2 [15, 16]. Also provide the abbreviation NRF2 since this is again later mentioned in the text.
We edited the text as suggested (lines 327-338).
9. Line 66 is also the first time the authors mention microbes. Here it is mentioned as if it was clear to the reader that microbes play an important role in OM. But from the first 3 sections this is unclear. Given the scope of the journal, and the rest of the review, it would be highly beneficial to already mention in the first sections that the oral microbiota plays a key role in the pathogenesis of OM; see suggestions above.
We edited the text as suggested (lines 50-68).
10. In this respect, section 4 can also consequently be adapted in this respect. Most of the preventive measures listed (Smoking, Alcohol, Denture use, low sugar and non acidic food, oral rinsing with
3
saline) will significantly impact the oral microbiota. But are currently simply listed without any context to why these preventive measures are important in shaping the microbiota. With respect to the previous comment, rewriting this more in the scope of the review will make the section look less like “patient information”.
We revised the text as suggested (lines 299-337).
11. Further, Line 66 mentions microbial overlap. What is specifically mentioned by this?
We removed this sentence in the revised version of the manuscript.
12. Tables 1 and2: need to be reformatted, because in some cases it is difficult to distinguish which effect belongs to each agent.
We edited the tables as indicated by the reviewer.
13. Lines 93-95 referring the tables should be combined to: Locally and systemically applied agents to treat OM are listed in Table 1 and Table 2 respectively.
We revised the text as suggested (lines 299-301).
14. That the therapies are listed from the most to the less commonly used should be mentioned in the table caption. For example: Table 1 title: Locally applied agents for the treatment of OM listed from most to less commonly used. Same for Table 2.
We revised the text as suggested (lines 299-301).
15. In section 5 Line 106 “mucosal barrier impairment and radiation-related dysbiosis favors the colonization and invasion of commensal bacteria and fungi.” Commensal microbes are by definition already colonizers of the mucosal surfaces, it should rather be stated that these overgrow and that the dysbalance of allows these normally harmless commensals to cause disease.
We amended as requested (page 4, line 143-151).
16. In lines 111-113 it is mentioned that Candida species are the main commensal yeast in the GI-tract. I would say that this information is redundant within the scope of this review.
We amended as requested.
17. In 114 the citation is missing.
We added the missing reference. Now is Ref.21 (line 171).
18. In lines 127-129 the information on how tobacco, antibiotics, oral prosthesis and poor oral hygiene contributes should be included in the section on preventive measures for oral mucositis.
We understand the reviewer’s suggestion, although we kept this information in the section 4, being related to risks of fungal infections and not of OM.
19. Lines 131-133 the clinical lesions due to fungal infection either need to be more clearly explained or their names not mentioned at all if they are not relevant. Knowing only the names of the lesions doesn’t add extra information.
4
We removed the name of the lesions (lines 192-194).
8. To better organize the flow of the review the oral microbiota section should be placed before the opportunistic fungal infection section. Furthermore, another example that the microbiota needs to be integrated in the first few paragraphs is that prophylaxis treatment is already mentioned in section 4 and is mentioned in the fungal infection section that dysbiosis leads to fungal infection. Such redundant overlaps (E.g. in line 215 microbial dysbiosis is explained but this explanation is already discussed above) and duplications can be avoided by thoroughly restructuring the review.
We changed the layout of the sections to improve the flow also accordingly to both the reviewers. We added several information about the oral microbiome in the first sections also avoiding overlaps.
20. Lines 217-220 consider whether the information about the eHOMD is relevant for the topic. If so, move information to line 208 where the database is discussed. Or only discuss the most recent database, which is most relevant.
We removed the eHOMD topic as suggested.
21. Figure 1b needs to be elaborated. Not a lot of information on the tissue parameters is given. For example, how is budding studied in these models?
5
We improved the figure as suggested and we decided to remove the focus on organoids to better emphasize the main topic.
The manuscript needs to be polished, as there are quite some grammar mistakes, spelling, and formatting inconsistencies. Se below my specific suggestions: Make sure consistently a space before the references is inserted Line 10 remove “highly” before “predispose” Line 14 remove “multiple” before “benefits” Line 16 remove “murine organoids” as only human organoids is discussed in this review Line 16-17 change order to: have already been exposed Line 19 change “host commensals” to: host-commensal Line 34 a new paragraph is started, but reading the text this looks unneccesary that shouldn’t be a new paragraph. Further rather place reference [7] at the end of the sentence or after “erythema”. Line 35 change “the first signs of OM (Figure 1).” To: the first visible sign of OM (Figure 1). Line 35-38 Consider rewriting the sentence “In the subsequent epithelial phase the damage of the membrane basement with loss of the protective barrier determines different degrees of desquamation and ulceration, which appears when the cumulative dose reaches 30 Gy [80].” To: Damage of the basement membrane beneath the epithelial cell layer leads to loss of the protective barrier which determines different degrees of desquamation and ulceration when the cumulative dose reaches 30 Gy. Line 41 Change “narcotic analgesics, prolonged hospitalization are” to: narcotic analgesics and prolonged hospitalization are Line 42 change “led” to: lead Line 48 change “do not” to: does not Line 51-52 change “appear.” to: appears Line 53-54 change “it is wide spreading used.” to: it is widely used. Line 60 change “oral hygiene, dental status” to: oral hygiene and dental status Line 68 First time the serial comma is used before “and” Line 68-69 change “xerostomia are linked with an increased bacterial colonization and severe OM [7]; consequently it is recommended their cure before the start of RT.” to xerostomia are linked with increased bacterial colonization and severe OM [7]; consequently their cure is recommended before the start of RT.” Line 86 Add by before “venous infusion” Line 89 Change “applied” to: application Line 92 Change “main cornerstones in the” to: main cornerstones of Line 99 Write: Opportunistic fungal infections instead of “fungal opportunistic infections” Line 113 replace “of” by: for Line 115-116 Radiotherapy could be abbreviated RT.
6
Lines 135-139: consider rewriting. This is a very long sentence that could be split in multiple. Line 142 write: main microbial risks instead of: “main microbial risk” Line 144 no need to split in two paragraphs. Line 144 write: to resist Candida dissemination instead of: “to resist to Candida dissemination” Line 148-149 Change “important and essential niche” to: an essential niche Lines 149-150 consider writing “In addition, the innate defenses with the highest importance in the oral cavity are basically related to…” Line 153 remove interestingly. Line 162 change “regulates” to: regulate Line 167 correct “contest” to: context Line 170 insert: in between “IL-17” and “both” Line 172 write: early and late proinflammatory response instead of “proinflammatory early and late response” Line 175 correct “dependend” to: dependent Lines 175-177 rewrite, because it says “it will be important to understand ... IL-17 immune signature importance”. Line 177 write: to resist Candida infection instead of “to resist to Candida infection” Line 178 keep spelling of “proinflammatory” consistent Line 180 change “may suggest” to suggests Line 181-182 Change sentence to read: it would be more essential to reconstitute the epithelial layers than to induce strong inflammation although it is antimicrobial Line 184 change the verb “influence” for one more specific. (e.g. epithelial proliferation increases/decreases the sucseptibility. Line 189 indicate that IL-22R is the IL-22 receptor Keep spelling of IL-22R consistent line 189 vs line 195 Correct placing of ref [52], [54], [55]. Line 206 correct “contest” to: context Line 206 change “in to” to: into (also correct this in other parts of the manuscript) Line 210 write limitations instead of “limitation” Lines 210-212 consider rewriting so as not to repeat information. Also use: culturable and unculturable instead of “cultivable” and “uncultivable” Line 212 correct spelling of “phila” to: phyla Line 220 remove the comma after “researchers” Line 226 insert: and before “TM7” Line 227 change “non smokers” to: non-smokers
7
Line 230 remove the capitalization of the P in “Gammaproteobacteria” Line 235 change “was not dissimilar from” to: was similar to Line 237 change reference formatting to: [61, 62] Line 247 insert:that before “positively correlated” Line 249 change “viceversa” to: vice versa and italicize Line 258 a new paragraph is not needed here and the word “Here” needs to be removed at the end of the sentence Line 261-262 “and” is italicized. Line 261-262 Remove etc., this does not provide any useful information Line 263 consider not starting a new paragraph. Line 270 consider changing “knowledge” for another alternative such as understanding. Line 272 change the expression “good combination” for something more specific, such as: beneficial combination. Line 281 write: fermenting instead of “ferment”. Line 282 repleace “very puzzling” by: complicated Line 292 change reference formatting to: [70, 71] Line 299 correct “in to” to: to Line 315 remove “The” Line 320 change “organoid” to: organoids Line 323 change “it is already under development the possibility” to: it is already under development with the possibility Lines 321-324 consider rewriting for more clarity and correct word order in the sentence. Lines 326-327 correct to: Microbes can also form biofilms or trigger antifungal innate immune responses
We thank the reviewer for the detailed revision. We have amended accordingly the text.

Reviewer 2 Report

In this manuscript, the authors attempt to review how radiation-induced oral mucositis relates to damage and secondary infections in the oral cavity. Overall the sections outlined in the current manuscript need to be expanded and more details provided. The topics mentioned in the abstract are only given brief attention in the text. The manuscript in its current form is not recommended for publication as explained in the following points:

  • Section 1. Radiation induced oral mucositis
    • This section is not cited properly. For example line 31-33.
    • The stages of OM should be described.
    • Sonis and colleagues should be referenced as it relates to grading scale, stages of OM, etc.
  • More details are needed for section 2 and 3, with proper citations.
  • Section 4. The bullet point format is very strange and not appropriate. Additionally, there are not enough details.
  • Section 5 Fungal opportunistic infections
    • While fungal infections are the most common in OM (and the focus in this review) there should be some mention of other infections occurring with head and neck irradiation. For example in line 105 dysbiosis is mentioned, but no clarification about why this is important.
  • Section 6
    • This section is very confusing. It is not clear how any of the antifungal mechanisms mentioned relates to HN OM-related infections. This is for the most part because very few details are provided.
  • Section 7
    • This section is disjointed. In previous sections, fungal infections were the focus and now bacterial flora is mentioned. There should be some continuity with previous sections.
    • The section title seems to be an afterthought in the text. A better title is warranted.
  • Section 7.2
    • It is not clear that the closing sentences have been addressed in this section. (line312-314). The topics reviewed in section 7.2 are a bit random and don’t relate back to OM.
  • Section 7.3 –
    • This section is not appropriate. The abstract makes the point of mentioning organoid systems, but then the text is not consistent with this. There are not enough details to understand the importance of this to OM.

It is not clear how the figure relates back to the topics covered in the manuscript. And the legend is not appropriately explained.

There are grammatical and spelling mistakes throughout the manuscript.

Author Response

Review# 1

In this manuscript, the authors attempt to review how radiation-induced oral mucositis relates to damage and secondary infections in the oral cavity. Overall the sections outlined in the current manuscript need to be expanded and more details provided. The topics mentioned in the abstract are only given brief attention in the text. The manuscript in its current form is not recommended for publication as explained in the following points:

We thank the reviewer for the suggestions. We have improved the different sections expanding the text and providing more details. Much more attention we put to improve the overall quality of the language used in the manuscript.

ď‚· Section 1. Radiation induced oral mucositis

This section is not cited properly. For example line 31-33.

We thank the reviewer for this suggestion. We have now provided additional references for the entire section (lines 31-68).

ď‚· The stages of OM should be described.

Sonis and colleagues should be referenced as it relates to grading scale, stages of OM, etc.

We have appreciated the suggestion and therefore we have better described the stages of OM (lines 30-49).

ď‚· More details are needed for section 2 and 3, with proper citations.

We have added more details as suggested (lines 69-136). piazzale Gambuli 1 06129 Perugia Teresa

ď‚· Section 4. The bullet point format is very strange and not appropriate. Additionally, there are not enough details.

We agree with the reviewer that the section was not exhaustive. Therefore, we have extensively changed this section and removed the bullet point format as suggested. In addition, we moved the section ahead to describe in the detail the risk of fungal infection before in the text.

ď‚· Section 5 Fungal opportunistic infections

While fungal infections are the most common in OM (and the focus in this review) there should be some mention of other infections occurring with head and neck irradiation. For example in line 105 dysbiosis is mentioned, but no clarification about why this is important.

We amended as requested this section (lines 137-208).

ď‚· Section 6

This section is very confusing. It is not clear how any of the antifungal mechanisms mentioned relates to HN OM-related infections. This is for the most part because very few details are provided.

We added more details to better explain how the scientific field of antifungal immunology needs to be better connected with HN OM-related infections (lines 209-295).

ď‚· Section 7

This section is disjointed. In previous sections, fungal infections were the focus and now bacterial flora is mentioned. There should be some continuity with previous sections.

We have now changed the order of the sections to give more continuity with previous sections as suggested.

ď‚· The section title seems to be an afterthought in the text. A better title is warranted. Section 7.2

We have now changed the title as suggested.

ď‚· It is not clear that the closing sentences have been addressed in this section. (line 312-314).

We have changed this section accordingly to reviewer’s suggestions.

ď‚· The topics reviewed in section 7.2 are a bit random and don’t relate back to OM.

We have changed this section accordingly to reviewer’s suggestions.

ď‚· Section 7.3 – This section is not appropriate. The abstract makes the point of mentioning organoid systems, but then the text is not consistent with this. There are not enough details to understand the importance of this to OM.

We removed this section in the new version of the manuscript as suggested. However, we think that this new technology is very important for the field. Thus, we mentioned this new advance in culture system in the section 8.

ď‚· It is not clear how the figure relates back to the topics covered in the manuscript. And the legend is not appropriately explained.

We agree with the reviewer and thus we have generally changed the figure layout.

ď‚· There are grammatical and spelling mistakes throughout the manuscript.

We have made revision of the text by mother tongue language expert to improve the quality of the of the text.